# MicroRNA-deficient mouse embryonic stem cells acquire a functional interferon response

Jeroen Witteveldt, Lisanne I Knol, Sara Macias*

Institute of Immunology and Infection Research, School of Biological Sciences, University of Edinburgh, Edinburgh, United Kingdom

**Abstract** When mammalian cells detect a viral infection, they initiate a type I interferon (IFNs) response as part of their innate immune system. This antiviral mechanism is conserved in virtually all cell types, except for embryonic stem cells (ESCs) and oocytes which are intrinsically incapable of producing IFNs. Despite the importance of the IFN response to fight viral infections, the mechanisms regulating this pathway during pluripotency are still unknown. Here we show that, in the absence of miRNAs, ESCs acquire an active IFN response. Proteomic analysis identified MAVS, a central component of the IFN pathway, to be actively silenced by miRNAs and responsible for suppressing IFN expression in ESCs. Furthermore, we show that knocking out a single miRNA, miR-673, restores the antiviral response in ESCs through MAVS regulation. Our findings suggest that the interaction between miR-673 and MAVS acts as a switch to suppress the antiviral IFN during pluripotency and present genetic approaches to enhance their antiviral immunity.
DOI: https://doi.org/10.7554/eLife.44171.001

## Introduction

Type I interferons (IFN) are crucial cytokines of the innate antiviral response. Although showing great variation, most mammalian cell types are capable of synthesizing type I IFNs in response to invading viruses and other pathogens. Once type I IFNs are secreted, they activate the JAK-STAT pathway and production of interferon-stimulated genes (ISGs) in both the infected and neighbouring cells to induce an antiviral state (*Ivashkiv and Donlin, 2015*). Two major signalling pathways are involved in IFN production in the context of viral infections. The dsRNA sensors RIG-I and MDA5 initiate a signalling cascade that signals through the central mitochondrial-associated factor MAVS, ultimately activating *Ifnb1* transcription. The cGAS/STING pathway is activated upon detection of viral or other foreign DNA molecules and uses a distinct signalling pathway involving the endoplasmic reticulum associated STING protein (*Chan and Gack, 2016*).

Despite its crucial function in fighting pathogens, pluripotent mammalian cells do not exhibit an IFN response. Both mouse and human embryonic stem cells (ESCs) (*Wang et al., 2013*; *Chen et al., 2010*) as well as embryonic carcinoma cells (*Burke et al., 1978*) fail to produce IFNs, suggesting that this function is acquired during differentiation. The rationale for silencing this response is not fully understood but it has been proposed that in their natural setting, ESCs are protected from viral infections by the trophoblast, which forms the outer layer of the blastocyst (*Delorme-Axford et al., 2014*). ESCs exhibit a mild response to exogenous IFNs, suggesting that during embryonic development, maternal IFN could have protective properties (*Hong and Carmichael, 2013*; *Wang et al., 2014*). In mouse ESCs, a Dicer-dependent RNA interference (RNAi) mechanism, reminiscent to that of plants and insects, is suggested to function as an alternative antiviral mechanism (*Maillard et al., 2013*). And in humans, ESCs intrinsically express high levels of a subgroup of ISGs in the absence of infection, bypassing the need for an antiviral IFN response (*Wu et al., 2018*; *Wu et al., 2012*). All

*For correspondence:
sara.maciasribela@ed.ac.uk

**Competing interests:** The authors declare that no competing interests exist.

**eLife digest** Living cells are under constant attack from disease-causing agents, such as viruses and bacteria. As a result, they have evolved various protective mechanisms to fight off these agents. One of the most important ways that an animal cell protects itself from infection is through the interferon response, which warns the cell of approaching viruses, prompting it to prepare to defend itself. Virtually all healthy cells have an active interferon response, except for stem cells, which have switched off this defensive mechanism, for unknown reasons. This makes stem cells more susceptible to infections.

Stem cells are specialized cells that play an essential role in developing the early embryo. The two defining characteristics of these cells – their ability to divide indefinitely, and develop into all cell types – offers great therapeutic potential, as they can be used to 'replace' damaged cells and tissues. However, without an interferon response, stem cells are likely to become infected when moved into a new environment, counteracting their therapeutic benefits. Now, Witteveldt et al. investigate how stem cells turn off this viral defence mechanism, and whether turning it back on will affect their ability to divide and form new tissues.

Using stem cells taken from the embryos of mice, Witteveldt et al. found that the interferon response is turned off by specific small molecules of RNA. These small RNA molecules block a protein in the pathway that recognizes viruses and activates a defence. Genetically engineering stem cells to be deficient in these small RNA molecules led to an increased resistance to viral infections. Importantly, modifying stem cells in this manner had no obvious impact on the characteristic traits that give stem cells their therapeutic potential.

Temporarily increasing the interferon response of stem cells as they are moved into a new environment could potentially make stem cell treatments more effective. However, more work is needed to investigate whether the same approach can be applied to human cells, and determine what negative effects may be associated with turning on the interferon response.

DOI: https://doi.org/10.7554/eLife.44171.002

these suggest that different antiviral pathways are employed depending on the differentiation status of the cell. Silencing of the IFN response during pluripotency may also be essential to avoid aberrant IFN production in response to retrotransposons and endogenous retroviral derived dsRNA, which are highly expressed during the early stages of embryonic development and oocytes (*Ahmad et al., 2018*; *Grow et al., 2015*; *Macia et al., 2015*; *Peaston et al., 2004*; *Macfarlan et al., 2012*). Furthermore, exposing cells to exogenous IFN induces differentiation and an anti-proliferative state, which would have catastrophic consequences during very early embryonic development (*Borden et al., 1982*; *Hertzog et al., 1994*).

All these observations support a model in which cells gain the ability to produce IFNs during differentiation. One particular class of regulatory factors that are essential for the successful differentiation of ESCs are miRNAs (*Greve et al., 2013*). These type of small RNAs originate from long precursor RNA molecules, which undergo two consecutive processing steps, one in the nucleus by the Microprocessor complex, followed by a DICER-mediated processing in the cytoplasm (*Treiber et al., 2018*). The Microprocessor complex is composed of the dsRNA binding protein DGCR8 and the RNase III DROSHA which are both essential for mature miRNA production (*Gregory et al., 2004*; *Lee et al., 2003*). In addition, mammalian DICER is also essential for production of siRNAs (*Bernstein et al., 2001*). The genetic ablation of *Dgcr8* or *Dicer* in mice blocks ESCs differentiation suggesting that miRNAs are an essential factor for this, as these are the common substrates for the two RNA processing factors (*Wang et al., 2007*; *Kanellopoulou et al., 2005*).

In this study, we show that miRNAs are responsible for suppressing the IFN response during pluripotency, specifically to immunostimulatory RNAs. We found that miRNA-deficient ESCs acquire an IFN-proficient state, are able to synthesize IFN-β and mount a functional antiviral response. Our results show that miRNAs specifically downregulate MAVS (mitochondrial antiviral signalling protein), an essential and central protein in the IFN response pathway. In agreement, ESCs with increased MAVS expression or knock-out of the MAVS-regulating miRNA miR-673, resulted in an increased IFN production and antiviral response. Our results support a model where the MAVS-miR-673

interaction acts as a switch to suppress the IFN response and consequently virus susceptibility during pluripotency.

## Results

### ESCs fail to express IFN-β in response to viral DNA/RNA

There are two major pathways for sensing intracellular viral infections and consequent activation of the IFN response in cells. One senses dsRNA, usually originating from RNA viruses, with MAVS as a central factor, and the second senses dsDNA, from DNA- and retroviruses signalling through STING (*McFadden et al., 2017*). It has been shown that mouse ESCs do not produce type I IFNs in response to poly(I:C) transfection, a synthetic analogue of dsRNA classically used to mimic viral RNA replication intermediates (*Wang et al., 2013*). In contrast, it is still unknown how mouse ESCs respond to immunostimulatory DNA. To study this, two different mouse ESC cell lines (ESC1 and ESC2) were transfected with poly(I:C) and G$_3$-YSD, an HIV-derived DNA that stimulates the cGAS/STING pathway (*Herzner et al., 2015*). As controls, NIH3T3 fibroblasts and BV-2 microglial cells were included. As expected, the transfection of poly(I:C) did not result in *Ifnb1* expression in both ESC lines (*Figure 1A*). ESCs also failed to activate *Ifnb1* expression upon G$_3$-YSD transfection, suggesting that the cGAS/STING pathway was also inactive (*Figure 1B*). Similarly, NIH3T3 cells, which have also been previously shown to have a defect in this specific pathway (*Cheng et al., 2018*), did not express *Ifnb1* in response to G$_3$-YSD (*Figure 1B*). These same cell lines were infected with the (+) ssRNA virus TMEV (Theiler's Murine Encephalomyelitis Virus) and showed that ESCs are at least 30 times more sensitive than NIH3T3 and BV-2 cells, which correlates with the ability of these cell lines to induce *Ifnb1*mRNA expression (*Figure 1C*).

The ability of cells to express IFN in response to viruses or immunogenic nucleic acids is assumed to be acquired during differentiation. To test this model, we *in vitro* differentiated both ESC lines with retinoic acid and determined their ability to respond to poly(I:C). Briefly, embryoid bodies were generated by a hanging droplet method for 48 hr before being cultured in the presence of retinoic acid for 2 or 10 days. Samples from each of these time points were analysed for expression of pluripotency and differentiation markers. The pluripotency markers *Nanog* and *Pou5f1* (*Oct4*) showed a rapid decrease in mRNA expression during differentiation in both the cell lines (*Figure 1—figure supplement 1A*), whereas differentiation markers *Neurog2*, *Gata6* and *Gata4* showed a gradual increase (*Figure 1—figure supplement 1B*) confirming successful differentiation of the ESCs. Next, we compared the ability of ESCs (day 0) and retinoic-acid differentiated cells after 10 days (day 10) to express *Ifnb1* mRNA in response to poly(I:C), and confirmed that differentiated cells acquired the ability to synthesize *Ifnb1* to similar levels to the positive control cell line, BV-2 (*Figure 1D*).

### Dicer-deficient ESCs acquire an active IFN response

Given the relevance of RNAi as an antiviral mechanism in mouse ESCs (*Maillard et al., 2013*), we next asked if ESCs, in the absence of the central factor for RNAi, ICER, would be more susceptible to RNA viruses. Unexpectedly, *Dicer*$^{-/-}$ ESCs were more resistant to viruses compared to their wild-type counterparts (previously named ESC2) (*Figure 2A*, left). Similar results were obtained using the (-) ssRNA virus, Influenza A (IAV) (*Figure 2A*, right). Importantly, mammalian Dicer has a dual function, being essential for both siRNA and miRNA biogenesis. To determine whether these differences in viral susceptibility were due to the activity of Dicer on siRNA or miRNA production, we compared *Dicer*$^{-/-}$ cells with ESCs lacking the essential nuclear factor for miRNA biogenesis, *Dgcr8*. The absence of *Dgcr8* also decreased TMEV and IAV viral susceptibility, suggesting that miRNAs are responsible for suppressing the antiviral response in ESCs (*Figure 2A*). Interestingly, *Dgcr8*$^{-/-}$ cells were more resistant to virus infection than *Dicer*$^{-/-}$ cells, which supports a dual function for DICER by also acting as a direct antiviral factor targeting viral transcripts for degradation by RNAi. To rule out the possibility of morphological differences influencing viral susceptibility, we performed a virus binding and entry assay which showed no differences (*Figure 2—figure supplement 1*).

Even though ESCs lack an IFN response, we wondered whether the differential resistance to viral infections were the result of abnormal IFN activation due to the absence of miRNAs. To test this hypothesis, we transfected the dsRNA analogue, poly(I:C) and the immunogenic G$_3$-YSD DNA in Dgcr8 or Dicer deficient mESCs, and quantified *Ifnb1* expression by RT-qPCR and ELISA. ESCs

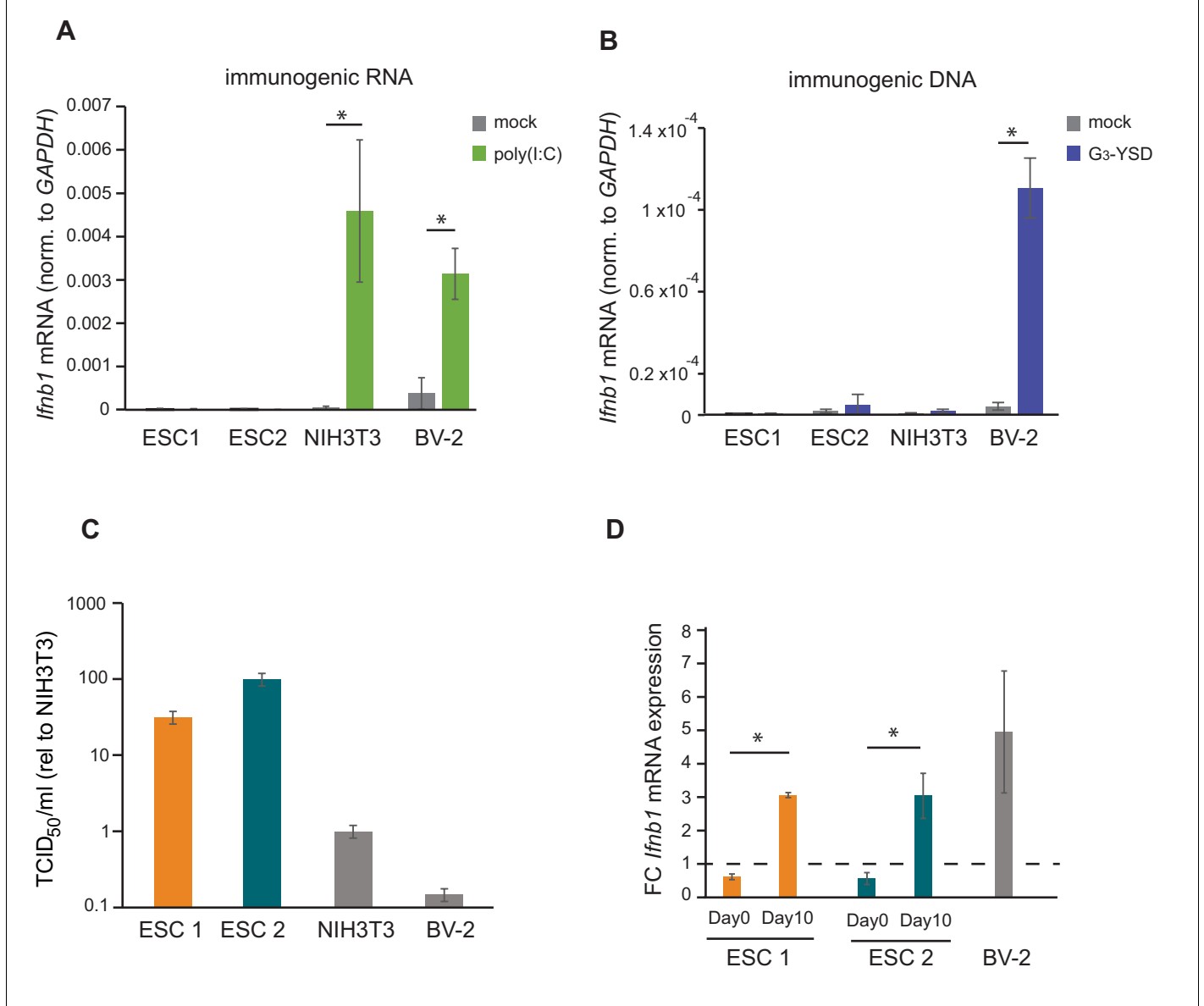

**Figure 1.** ESCs lack IFN response and are more susceptible to viral infection. (a) Quantification of *Ifnb1* expression in ESCs and the somatic mouse cell lines NIH3T3 and BV-2 after transfection with the dsRNA analogue poly(I:C). Data show the average (n = 3)±s.e.m, (*) p-value<0.05 by t-test. (b) Quantification of *Ifnb1* expression after activation of the cGAS response by Y-DNA (G3-YSD) in the same cells lines as (a). Data show the average (n = 3, except for ESC2, n = 2)±s.e.m, (*) p-value<0.05 by t-test. (c) Susceptibility (TCID$_{50}$/ml) of same cell lines as used in (a) to TMEV infection. (d) Quantification of *Ifnb1* expression in pluripotent and differentiated ESCs after activation with poly(I:C). Data show the average (n = 3) fold change over mock treated cells,±s.d. (*) p-value<0.05 by t-test.
DOI: https://doi.org/10.7554/eLife.44171.003

The following figure supplement is available for figure 1:

**Figure supplement 1.** Retinoic acid differentiation of ESCs.
DOI: https://doi.org/10.7554/eLife.44171.004

lacking miRNAs (*Dgcr8*[-/-] or *Dicer*[-/-]) were able to respond to the dsRNA analogue, poly(I:C) and express *Ifnb1* mRNA and protein in a dose dependent manner (*Figure 2B* and *Figure 2—figure supplement 2A–C*), whereas no significant response was observed with immunostimulatory DNA (*Figure 2B*). These results show there is a correlation between viral susceptibility and the ability of miRNA-deficient ESCs to express IFN-β, and suggest that miRNAs are responsible for silencing the IFN response to dsRNA. To further establish the involvement of IFN expression in these

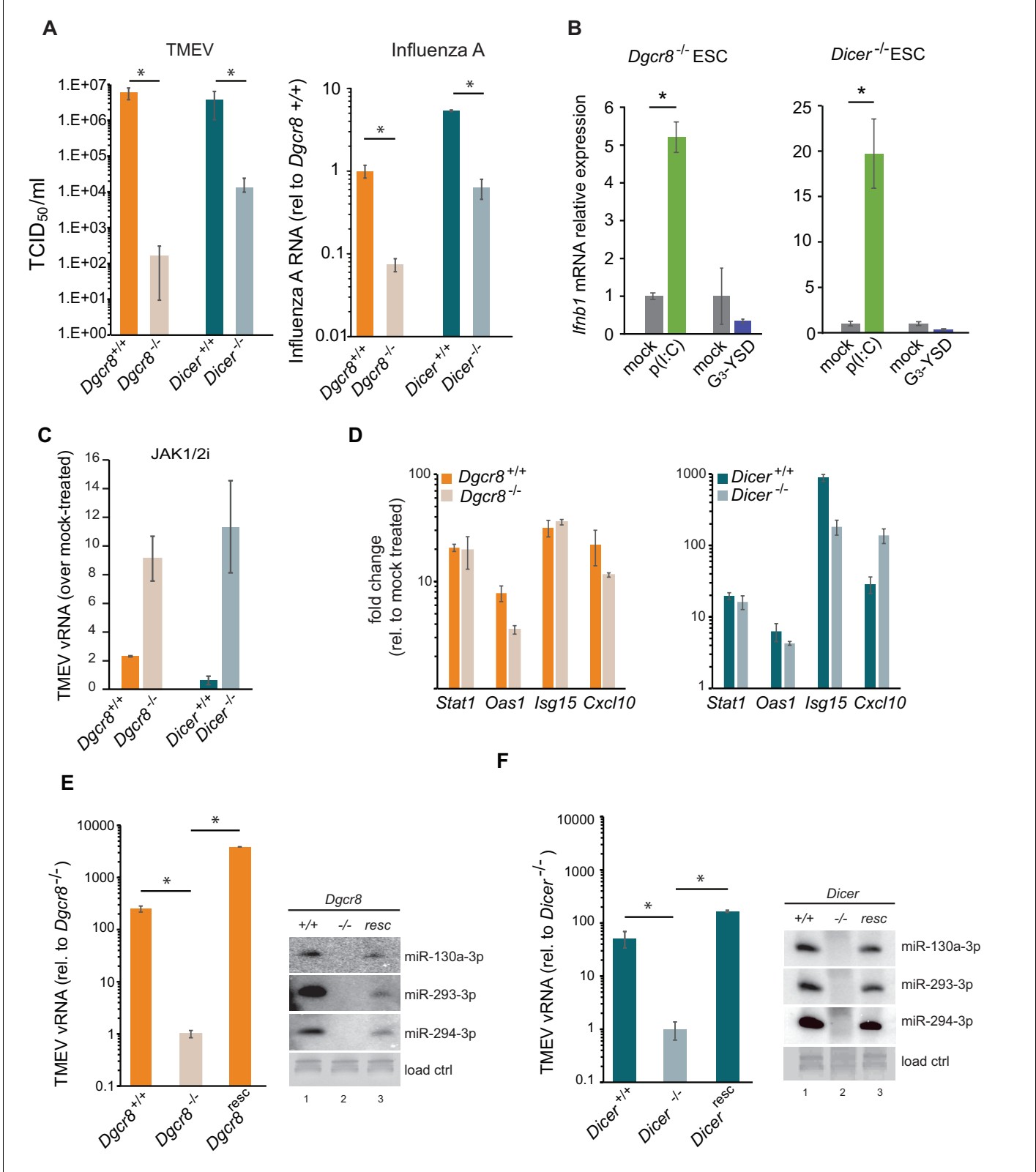

**Figure 2.** MiRNAs regulate IFN response. (a) (left) Susceptibility (TCID$_{50}$/ml) of miRNA deficient cells (*Dgcr8$^{-/-}$*, *Dicer$^{-/-}$*) and wild-type parental cells (*Dgcr8$^{+/+}$*(ESC1), *Dicer$^{+/+}$*(ESC2)) to TMEV infection, higher values represent higher susceptibility (n = 4, p-value<0.05, t-test). (right) Quantification of Influenza A replication after infection of the same cell lines, data show the average (n = 3)±s.d. (*) p-value<0.05 by t-test. (b) Quantification of *Ifnb1* expression of ESCs lacking *Dgcr8* or *Dicer* to stimulation with poly(I:C) and G$_3$-YSD. Data show average (n = 3)±s.d., normalized to mock, (*)
*Figure 2 continued on next page*

Figure 2 continued

p-value<0.05 by t-test. (c) Quantification of TMEV vRNA upon JAK1/2 inhibition by Ruxolitinib treatment. Data show the average (n = 3)±s.d. (d) qRT-PCR analyses of ISGs expression after stimulation of wild-type and miRNA-deficient ESCs with IFN-β. Data show average (n = 3)±s.d., normalized to mock treated cells (e, f) Quantification of TMEV replication after infection in *Dgcr8* (e) and *Dicer* (f) parental (+/+), deficient (-/-) and rescued (resc) cell lines. Data are normalized to miRNA-deficient cell lines susceptibility. Data show the average (n = 3)±s.d (*) p-value<0.05 by t-test. Northern blots for three stem-cell specific miRNAs, as control for knock-out and rescue of *Dgcr8* and *Dicer*, are shown at the right of each panel.
DOI: https://doi.org/10.7554/eLife.44171.005

The following figure supplements are available for figure 2:

**Figure supplement 1.** Viral entry in miRNA-deficient and wild-type ESCs.
DOI: https://doi.org/10.7554/eLife.44171.006

**Figure supplement 2.** miRNA-deficient ESCs express IFN in response to dsRNA.
DOI: https://doi.org/10.7554/eLife.44171.007

observations, we blocked IFN signalling using the JAK1/2 inhibitor Ruxolitinib before infecting cells with TMEV. As a result we observed no, or a very mild increase in TMEV viral replication in wild-type ESCs, but a significant increase in viral replication in miRNA-deficient ESCs (*Figure 2C* and *Figure 2—figure supplement 2D*). ESCs were also stimulated with exogenous IFN-β and confirmed that mouse ESCs retain the ability to respond to external IFNs, and, importantly, that miRNA deficiency did not alter ISG expression levels, supporting the hypothesis that the miRNA-mediated silencing of the IFN pathway in ESCs occurs upstream of IFN production (*Figure 2D*). To verify that the observed results are solely due to the absence of miRNAs, we rescued the knock-out cell lines by reintroducing Dgcr8 and Dicer and observed that these reverted to wild-type viral replication and susceptibility levels (*Figure 2E,F* and *Figure 2—figure supplement 2E*). As a control, we confirmed rescue of miRNA production by Northern blot (*Figure 2E,F*).

## miRNAs suppress MAVS expression in ESCs

To understand where the IFN pathway is silenced in ESCs we blocked the interferon response at defined points in the pathway and measured viral susceptibility. The inhibitor BX795 blocks TBK1/IKKε phosphorylation and consequently IRF3 transcriptional activity, whereas BMS345541 is an inhibitor of the catalytic subunits of IKK and thus blocks Nf-κB-driven transcription. Both transcription factors are essential for the expression of *Ifnb1* and other pro-inflammatory cytokines and initiation of an antiviral response (*Lawrence, 2009*; *Schafer et al., 1998*). Both inhibitors increased viral susceptibility in wild-type cells lines, however, the effect was far greater in the knock-out cell lines (*Figure 3A,B* and *Figure 3—figure supplement 1A–C*), suggesting that miRNAs regulate the interferon pathway upstream *Ifnb1* transcription.

We next aimed to identify the mechanism by which miRNAs silence IFN expression in ESCs, and analysed the proteomes of *Dgcr8⁻/⁻* and the rescued cell line by mass spectrometry. STRING analyses of the expression profiles revealed significant differences in a number of pathways, including ribosome structure/function, mitochondrial activity and the oxidative phosphorylation pathway, which were downregulated in the absence of miRNAs (*Figure 3C*, for complete list see *Figure 3—source data 1*). Measurement of Rhodamine 123 uptake in mitochondria, as an indirect measure for oxidative phosphorylation activity (*Scaduto and Grotyohann, 1999*), confirmed lower oxidative phosphorylation activity in the absence of miRNAs (*Dgcr8⁻/⁻* and *Dicer⁻/⁻*) (*Figure 3—figure supplement 1D*). A search for differentially expressed proteins involved in the IFN response did not reveal any significant changes except for the Mitochondrial antiviral-signalling protein (MAVS), which in contrast to many other mitochondria-related proteins, was upregulated in the absence of miRNAs. This protein has a central role in the RLR-induced (Rig-I-like receptors) IFN pathway, where activated MDA5 and RIG-I receptors translocate to the mitochondria and bind MAVS to ultimately induce Ifnb1 expression (*Kawai et al., 2005*). Western blot and qRT-PCR analysis confirmed that MAVS was the only factor consistently expressed to higher levels in both miRNA-deficient cell lines, *Dgcr8⁻/⁻* and *Dicer⁻/⁻* (*Figure 3D*, lanes 2 and 5, and *Figure 3—figure supplement 1E*), compared to a panel of other components of the same innate immune response pathway (*Figure 3—figure supplement 1F*).

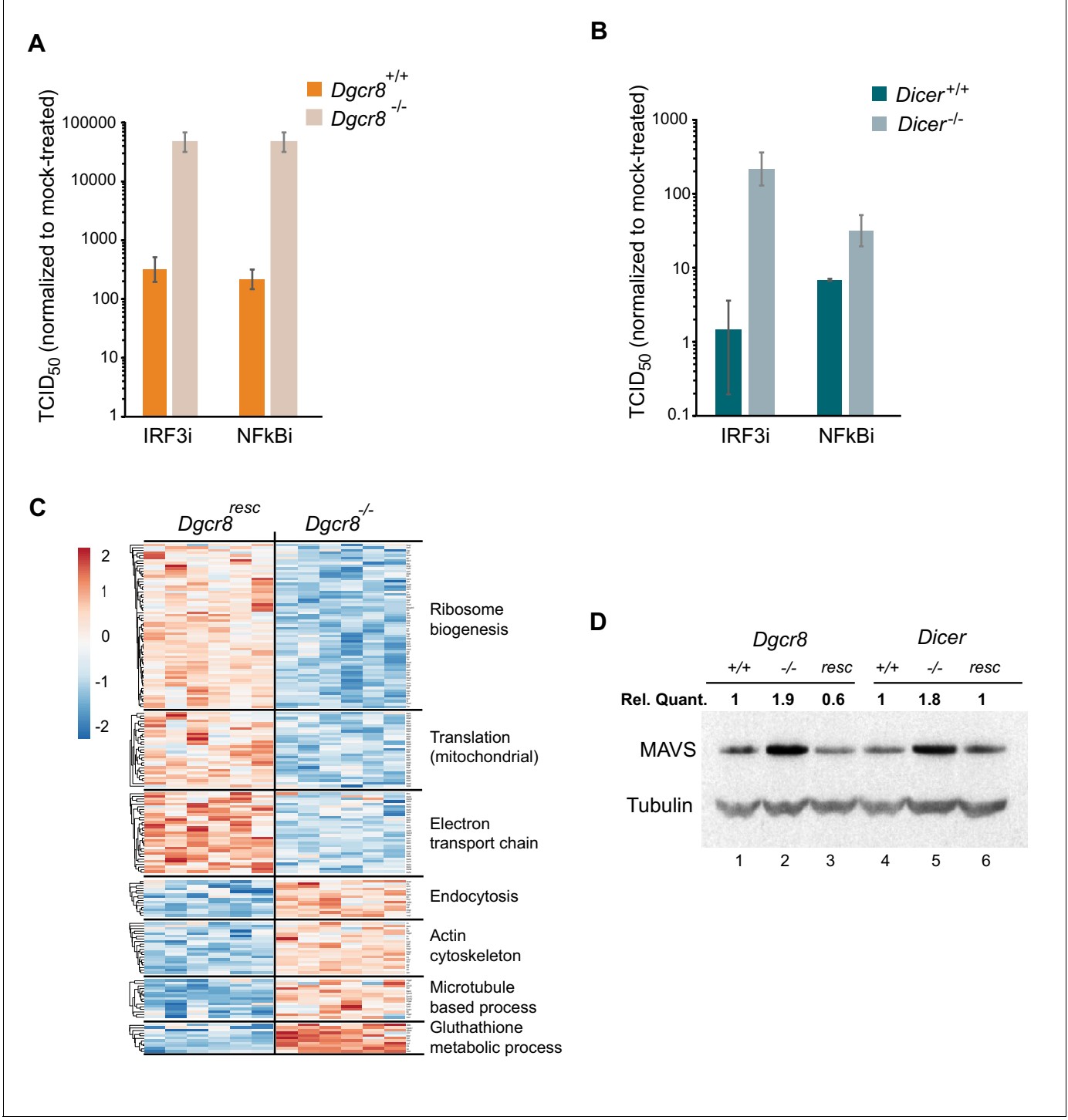

**Figure 3.** MAVS is downregulated by miRNAs in ESCs. (a, b) Susceptibility of *Dgcr8*[-/-], *Dicer*[-/-] and parental cells to TMEV infection after inhibition of IRF3 (BX795) (a) and Nf-κB (BMS345541) (b), normalized to mock-treated cells. (c) Heat map of significantly differentially expressed proteins (p<0.05) in the absence (*Dgcr8*[-/-]) or presence (*Dgcr8*[resc]) of miRNAs identified by STRING analysis. (d) Western blot analysis of MAVS expression in miRNA-deficient cells (*Dgcr8*[-/-] and *Dicer*[-/-], lanes 2 and 5), wild-type counterparts (*Dgcr8*[+/+] and *Dicer*[+/+], lanes 1 and 4) and respective rescued ESCs lines (*Dgcr8*[resc] and *Dicer*[resc], lanes 3 and 6). MAVS quantification normalized to Tubulin and relative to wild-type levels is shown at the top of the panel.
DOI: https://doi.org/10.7554/eLife.44171.008

The following source data and figure supplement are available for figure 3:

**Source data 1.** Source data of mass spectrometry results.
DOI: https://doi.org/10.7554/eLife.44171.010

*Figure 3 continued on next page*

*Figure 3 continued*

**Figure supplement 1.** miRNAs regulate MAVS and mitochondrial activity.

DOI: https://doi.org/10.7554/eLife.44171.009

## MAVS acts as a switch for IFN expression

To confirm the involvement of miRNAs on MAVS expression, a dual luciferase assay system was used where the 3'UTRs of *Mavs*, *Mda5* and *Rig-I* were fused to a luciferase reporter gene to compare luciferase activity in wild-type and knock-out ESCs. Only the *Mavs* 3'UTR showed relatively higher luciferase expression levels in the knock-out lines when compared to the empty plasmid, suggesting that the 3'UTR of *Mavs* is strongly regulated by miRNAs in ESCs (*Figure 4A*). For this reason, a miRNA-resistant form of *Mavs*, lacking its natural 3'UTR, was overexpressed in wild-type ESCs and infected with TMEV to test if cells regain viral resistance similar to miRNA deficient ESCs (*Figure 4B*). A 15-fold decrease in $TCID_{50}$ and significant reduction in vRNA levels were found compared to wild-type ESCs (*Figure 4C*). MAVS overexpressing cells also regained the ability to produce *Ifnb1* after stimulation with poly(I:C) (*Figure 4D*). All these experiments show that MAVS is a crucial target for the absence of the IFN response in ESCs.

## miR-673 is crucial to suppress antiviral immunity in ESCs

We next aimed to identify the miRNA(s) responsible for the regulation of MAVS in ESCs and selected a number of miRNA candidates based on literature, prediction software and public miRNA expression databases for further investigations. Previous experimental evidence has shown that human MAVS is regulated by miR-125a, miR-125b and miR-22 (*Hsu et al., 2017*; *Wan et al., 2016*). However, only miR-125a-5p and miR-125b-5p have conserved binding sites in mouse MAVS. Two additional miRNAs, miR-185–5p and miR-673–5p, were selected based on their DICER and DGCR8-dependent biosynthesis pathway, their high expression levels in mouse ESCs and number of predicted binding sites in the *Mavs* 3'UTR (*Tang et al., 2006*; *Babiarz et al., 2008*). We transfected *Dgcr8*[-/-] cells with mimics of these miRNAs and measured *Mavs* mRNA and protein levels by RT-qPCR and western blot, respectively. Results showed reductions in MAVS protein and mRNA levels for all tested miRNAs (*Figure 5A* and *Figure 5—figure supplement 1A*). The infection of miRNA-transfected *Dgcr8*[-/-] cells with TMEV resulted in an increase in both susceptibility and viral replication for miR-125a-5p, miR-125b-5p and miR-673–5p, which correlated with the ability of these miRNAs to downregulate MAVS protein levels (*Figure 5B* and *Figure 5—figure supplement 1B*).

As an alternative approach, *Dgcr8*[+/+] cells were transfected with inhibitors to miRNAs miR-125a-5p, miR-125b-5p and miR-673–5p. Western blot analysis showed a clear increase in MAVS protein expression, especially for anti-miR-673–5p (*Figure 5C*). Because miR-673–5p showed the largest effect on MAVS protein expression both when depleted and overexpressed, we hypothesize that miR-673 is a crucial miRNA involved on MAVS regulation.

We further investigated the role of miR-673–5p in ESCs by creating stable knock-out cell lines for this miRNA by CRISPR/Cas9. Three cell lines were selected based on the genomic deletion and confirmed undetectable expression of miR-673–5p (*Figure 5—figure supplement 2A,B*). The absence of miR-673–5p was enough to observe an increase in MAVS expression both at the mRNA and protein levels (*Figure 5D* and *Figure 5—figure supplement 2C*). In addition, we measured miR-673 and MAVS expression levels in the mouse fibroblasts cell line, NIH3T3, which is proficient in producing IFN in response to dsRNA. Mouse fibroblasts had no detectable miR-673–5p, and MAVS protein expression was comparable to miRNA-deficient ESC (*Figure 5D* and *Figure 5—figure supplement 2B*), highlighting the correlation of MAVS expression with the ability of cells to activate *Ifnb1* expression in response to immunogenic RNA.

Next, miR-673-deficient cell lines were tested for TMEV susceptibility, which showed a consistent decrease in virus replication, similar to that observed in the absence of all miRNAs (*Dgcr8*[-/-]), suggesting this miRNA is essential in regulating the innate antiviral response in ESCs (*Figure 5E*). To test the relevance of IFNs on the increased antiviral resistance of miR-673[-/-] cell lines, we compared their sensitivity to TMEV infections in the presence of the JAK1/2 inhibitor, Ruxolitinib. Whereas inhibition of IFN signalling did not significantly increase the accumulation of viral RNA in wild-type ESCs (*Dgcr8*[+/+]), both miR-673-deficient and *Dgcr8*[-/-] ESCs showed a significant increase in viral RNA

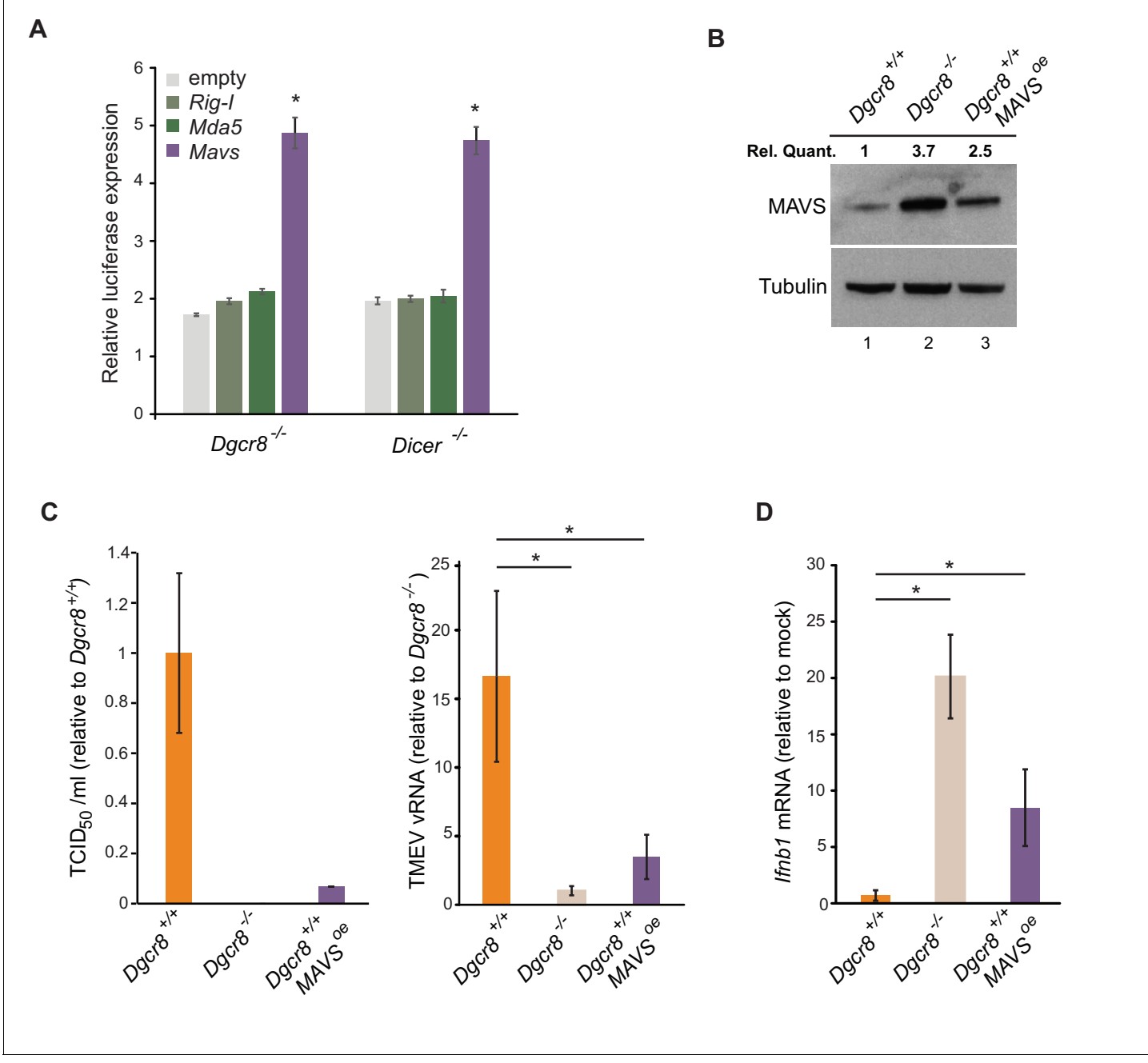

**Figure 4.** ESCs regain *Ifnb1* expression after MAVS overexpression. (a) Dual luciferase assay with *Mavs*, *Rig-I* and *Mda5* 3'UTRs in miRNA-deficient cells lines (*Dgcr8*$^{-/-}$ and *Dicer*$^{-/-}$). Data show the average (n = 3)±s.d normalized to Renilla and relative to the parental lines, (*) p-value<0.05 by t-test (b) Western blot of cell line overexpressing MAVS lacking the 3'UTR in *Dgcr8*$^{+/+}$ cells (lane 3). MAVS quantification normalized to Tubulin and relative to wild-type is shown at the top (c) Susceptibility (TCID$_{50}$/ml) of same cells lines as in (b) to TMEV infection (left panel) and quantification of viral RNA after TMEV infections in the same cell lines (right panel). Data show the average (n = 5)±s.d. (*) p-value<0.05 by t-test (d) *Ifnb* mRNA expression after poly(I: C) transfection of the same cell lines as in (b), average is represented (n = 3)±s.d, normalized to *Dgcr8*$^{+/+}$ cell line, (*) p-value<0.05 by t-test.
DOI: https://doi.org/10.7554/eLife.44171.011

upon treatment, confirming the role of miR-673 and IFNs in viral susceptibility (*Figure 5F* and *Figure 5—figure supplement 3A*). As controls, we confirmed that knocking out miR-673–5p expression did not affect the ability of ESCs to proliferate or induced spontaneous differentiation, which could cause differences in viral susceptibility (*Figure 5—figure supplement 3B–E*). Interestingly, during

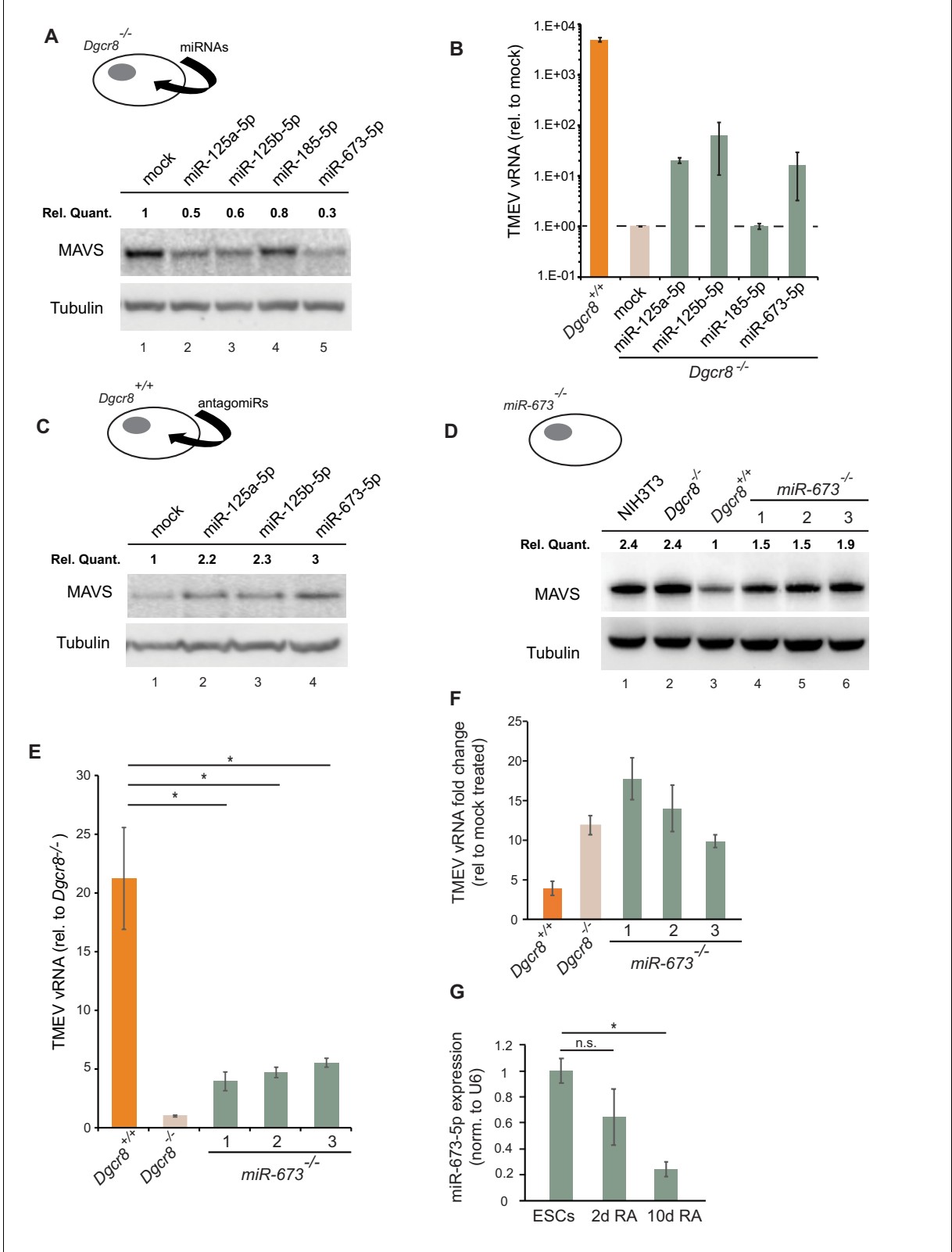

**Figure 5.** MiR-673–5p regulates MAVS. (a) Transfection of miRNA mimics miR-125a-5p, miR-125b-5p, miR-185–5p and miR-673–5p in *Dgcr8*[−/−] cells followed by MAVS western blot. MAVS quantification normalized to Tubulin and relative to wild-type is shown at the top (b) Quantification of TMEV replication by qRT-PCR in the same cell lines as in (a) (n = 3) (c) Western blot analysis of MAVS expression in *Dgcr8*[+/+] cells transfected with antagomirs against miR-125a-5p, miR-125b-5p and miR-673–5p. MAVS quantification normalized to Tubulin and relative to wild-type is shown at the top (d)

*Figure 5 continued on next page*

*Figure 5 continued*

Western blot analysis of MAVS expression in *miR-673^-/-* cell lines. MAVS quantification normalized to Tubulin and relative to wild-type is shown at the top (e) Quantification of TMEV vRNA in *miR-673* CRISPR knock-out cell lines in a *Dgcr8^+/+* background. Data show the average (n = 3)±s.d (*) p-value<0.05 by t-test. (f) Quantification of TMEV vRNA shown as fold-change compared to mock upon JAK1/2 inhibition by Ruxolitinib treatment. Data show the average (n = 3)±s.d (g) qRT-PCR quantification of mmu-miR-673–5p expression after retinoic acid (RA) differentiation of ESC. Data show the average (n = 2)±s.e.m, normalized to U6 snRNA, (*) p-val <0.05 by t-test.

DOI: https://doi.org/10.7554/eLife.44171.012

The following figure supplements are available for figure 5:

**Figure supplement 1.** Characterization of MAVS expression in ESCs and regulation by miRNAs.

DOI: https://doi.org/10.7554/eLife.44171.013

**Figure supplement 2.** Characterization of *miR-673^-/-* mouse embryonic stem cells.

DOI: https://doi.org/10.7554/eLife.44171.014

**Figure supplement 3.** Characterization of *miR-673 ^-/-* mouse ESCs.

DOI: https://doi.org/10.7554/eLife.44171.015

ESC differentiation with retinoic acid, expression of miR-673–5p became silenced, confirming previous results obtained with alternative differentiation protocols (*Knelangen et al., 2011*; *Zhao et al., 2014*; *Hadjimichael et al., 2016*; *Yang et al., 2016*), and suggesting that the expression levels of this miRNA negatively correlate with the ability of cells to activate the IFN response (*Figure 5G*).

Collectively, these data show that the IFN response in mouse ESCs is silenced by the post-transcriptional control of *Mavs* expression by miR-673–5p.

## Discussion

Several studies suggest that the pluripotent state of a cell is incompatible with an active IFN response (*Guo et al., 2015*). Both mouse and human stem cells fail to synthesize IFNs in response to dsRNA (*Wang et al., 2013*; *Chen et al., 2010*), implying that this characteristic is acquired during differentiation (*D'Angelo et al., 2016*). Embryonic carcinoma cells, which are still pluripotent, also fail to produce IFNs in response to viral RNA mimics (*Burke et al., 1978*). In agreement, reprogramming of somatic cells to iPSCs (induced pluripotent stem cells) leads to a loss of IFN response, suggesting the presence of regulatory mechanisms able to switch this antiviral pathway on or off between the differentiated and pluripotent states (*Chen et al., 2012*). Another feature of pluripotent cells is their attenuated response to exogenous type I IFNs. Mammalian pluripotent stem cells, iPSCs and embryonic carcinoma cells exhibit an attenuated production of ISGs upon type I IFN stimulation (*Hong and Carmichael, 2013*; *Irudayam et al., 2015*; *Wang et al., 2014*; *Burke et al., 1978*). Why these activities are supressed is still not understood, but it has been hypothesized that type I IFN stimulation could impair their self-renewal capacity, since these compounds are well-known antiproliferative agents and inducers of cell death (*Bekisz et al., 2010*). Indeed, type I IFNs are capable of inhibiting tumour cell division *in vitro* and are currently employed as an adjuvant to treat several types of cancers, acting as stimulants of the innate immune cellular response (*Bracci et al., 2017*).

Mouse ESCs express low levels of the RNA sensors TLR3, MDA5 and RIG-I, which could explain their inability to respond to dsRNA although no functional studies support this model so far (*Wang et al., 2013*). Our data shows an alternative scenario in which MAVS is the key factor for controlling the IFN response. The overexpression of a miRNA-resistant form of MAVS in wild-type ESCs is enough to enable dsRNA-mediated IFN activation, suggesting that dsRNA sensing is not a limiting step in the IFN pathway in ESCs. Regulation of MAVS alone proves to be an efficient mechanism to block dsRNA induced IFN expression compared to suppressing individual dsRNA sensors.

The observation that miRNAs only suppress RNA-mediated IFN activation, but not the DNA-mediated pathway, leads us to speculate about the reasons for silencing this specific response during pluripotency. Embryonic stem cells, and also earlier stages of embryonic development are characterized by high expression levels of specific retrotransposons (non-LTR) and endogenous retroviruses (LTR), which are a hallmark of their pluripotent state. This is in contrast to most somatic cell types that silence their expression (*Yin et al., 2018*). These repetitive elements produce cytoplasmic RNA molecules as an intermediate for mobilisation, which can be accidentally recognised as immunogenic or non-self RNAs, as it has been previously shown for the human non-LTR

retroelement Alu in the context of Aicardi-Goutires syndrome or for endogenous retroviruses (*Ahmad et al., 2018*; *Chiappinelli et al., 2015*; *Roulois et al., 2015*). Therefore, silencing the RNA-mediated IFN response during pluripotency would act as a protective mechanism for aberrant IFN activation by transposon-derived transcripts.

Cells that are incapable of activating the RNA-mediated IFN response have developed alternative antiviral defence pathways. The endonuclease DICER can act as an antiviral factor in mouse ESCs by generating antiviral siRNAs (*Maillard et al., 2013*). Detection of antiviral DICER activity is facilitated in the absence of a competent IFN response, such as in the case of pluripotent cells, but also in somatic cells where the type I IFN response has been genetically impaired (*Maillard et al., 2016*). These findings are supported by the observation that in IFN-competent cells, the RNA sensor LGP2 acts as an inhibitor of DICER cleavage activity on dsRNA (*van der Veen et al., 2018*). However, DICER activity has also been reported in other cell lines, independently of their IFN-proficiency capacity (*Li et al., 2016*). Interestingly, when we disrupt *Dicer* in ESCs, which inherently lack an IFN response and would theoretically render these cells highly sensitive to viral infections, they become more resistant by acquiring an active IFN response. All these results support the presence of extensive cross-talk between the different antiviral strategies, and suggests that cells have developed mechanisms to compensate for the loss of a specific antiviral pathway.

Our model shows that MAVS and miR-673 levels are the key factors regulating the IFN response to dsRNAs during pluripotency. Accordingly, overexpressing MAVS or knocking-out this single miRNA in ESCs is enough to enhance their antiviral response. Interestingly, this miRNA is only conserved in rodents, despite human ESCs also suppressing type I IFNs expression (*Hong and Carmichael, 2013*). This suggests that either other miRNAs regulate MAVS expression in human ESCs, or alternative mechanisms operate to silence IFN. Interestingly, human and mouse ESCs have been suggested to constitutively express a subset of ISGs to protect them from viruses (*Wu et al., 2018*). Our proteomics data suggest that, from all the ISGs detected, miRNAs did not significantly affect production of these antiviral factors, such as IFITM1, IFTIM2, IFITM3 amongst others. We have shown that engineering ESCs to acquire a functional IFN response significantly increases their antiviral immunity, highlighting the powerful antiviral effects of IFNs even during pluripotency.

Previous findings also support a general role for DICER and miRNAs acting as negative regulators of the IFN response in human and mouse models outside pluripotency (*Papadopoulou et al., 2012*; *Witteveldt et al., 2018*). In agreement, an indirect approach to deplete cellular miRNAs, by overexpressing the viral protein VP55 from Vaccinia virus, showed that miRNAs are also relevant to control the expression of pro-inflammatory cytokines during viral chronic infections, but not in the acute antiviral response (*Aguado et al., 2015*). However, the concept of miRNAs acting as direct antiviral factors is still controversial. It is relevant to mention that some of the results leading to this conclusion have been primarily generated in *DICER1*[-/-] HEK293T human cell line (*Bogerd et al., 2014*; *Tsai et al., 2018*), which has an attenuated IFN response due to low PRRs expression (*Rice et al., 2014*; *Witteveldt et al., 2018*).

We have shown that overexpression of MAVS or silencing specific miRNAs in a transient or stable manner improves the antiviral response of ESCs. These findings are the basis to further study the conservation of the miRNA-mediated regulation of the IFN response in somatic cells and in the context of human pluripotency. All these investigations will provide a deeper understanding and tool set on how to enhance the innate immunity of ESCs and their differentiated progeny, an especially relevant aspect in clinical applications.

## Materials and methods

### Cells and viruses

*Dgcr8* knockout (*Dgcr8*[-/-]) mouse ESCs were purchased from Novus Biologicals (NBA1-19349) and the parental strain, v6.5 (*Dgcr8*[+/+], also named in the text ESC1) from ThermoFisher (MES1402). *Dicer* [flox/flox] (*Dicer*[+/+], also named ESC2) and *Dicer* knockout (*Dicer*[-/-]) mouse ESCs were provided by R. Blelloch lab (University of California, San Francisco). All mESC cells were cultured in Dulbecco's modified Eagle Medium (DMEM, ThermoFisher) supplemented with 15% heat-inactivated foetal calf serum (ThermoFisher), 100 U/ml penicillin, 100 μg/ml streptomycin (ThermoFisher), 1X Minimal essential amino acids (ThermoFisher), 2 mM L-glutamine, $10^3$ U/ml of LIF (Stemcell Technologies)

and 50 µM 2-mercaptoethanol (ThermoFisher). Cells were grown on plates coated with 0.1% Gelatine (Embryomax, Millipore), detached using 0.05% Trypsin (ThermoFisher) and incubated at 5% $CO_2$ at 37°C. MDCK, BHK-21, BV-2 and RAW264.7 cells were cultured in Dulbecco's modified Eagle Medium (DMEM, ThermoFisher) supplemented with 10% heat-inactivated foetal calf serum (ThermoFisher), 100 U/ml penicillin, 100 µg/ml streptomycin (ThermoFisher), 2 mM L-glutamine and incubated at 5% $CO_2$ at 37°C. NIH3T3 cell line was provided by A. Buck lab, and grown in DMEM supplemented with 10% FCS. To determine cell proliferation kinetics, cells were seeded in 6-well plates at a density of $2*10^5$ cells/well. Cell numbers were determined in triplicates after 12, 24, 36 and 48 hr using a CASY cell counter.

Stocks of TMEV strain GDVII were grown on BHK-21 cells and frozen in aliquots at −80°C. Stocks of Influenza A virus strain PR8 (kindly provided by P. Digard, University of Edinburgh) were grown on MDCK cells in the absence of serum and in the presence of 2 µg/ml TPCK-treated trypsin and frozen in aliquots at −80°C. For TMEV infections, cells were infected for 1 hr with the required dilution, followed by replacement with fresh medium and incubation for the desired time. For the 50% Tissue Culture Infective dose ($TCID_{50}$) assays, seven serial dilutions of TMEV were prepared and at least six wells (in 96-well format) per dilution were infected and incubated for at least 24 hr before counting infected wells. $TCID_{50}$ values were calculated using the Spearman and Kärber algorithm. Influenza A virus infections were performed by infecting cells in the absence of serum for 45 min with the addition of 2 µg/ml TPCK-treated trypsin. After replacement of the inoculum with fresh serum containing medium the cells were incubated for the desired period.

## Differentiation of mESCs

To differentiate mESCs, they were first cultured as hanging droplets to induce embryoid body formation. For this, a single-cell suspension of $5 \times 10^5$ cells/ml was prepared in medium without LIF and 20 µl drops are pipetted on the inside of the lid of a 10 cm petri dish and hung upside-down. The petri-dish was filled with PBS to prevent drying of the hanging drops and incubated at 37°C, 5% $CO_2$ for 48 hr. The embryoid bodies were consequently washed from the lids and transferred to petri dishes to further differentiate, all in the absence of LIF. After another incubation time of 48 hr, medium was removed and replaced with fresh medium containing 250 nM of retinoic acid (Sigma-Aldrich) and incubated for 7 days while replacing the medium every 48 hr. After this incubation time, the embryoid bodies were collected and plated on normal gelatine-coated cell culture plates which allowed the embryoid bodies to adhere to the plastic and the cells to migrate from the embryoid bodies. Again, the medium was refreshed every 48 hr for the cells to further differentiate.

## Northern blot for miRNAs

Total RNA (15 µg) was loaded on a 10% TBE-UREA gel. After electrophoresis, gel was stained with SYBR gold for visualization of equal loading. Gel was transferred onto a positively charged Nylon membrane for 1 hr at 250 mA. After UV-crosslinking, the membrane was pre-hybridized for 4 hr at 40°C in 1xSSC, 1%SDS (w/v) and 100 mg/ml single-stranded DNA (Sigma). Radioactively labelled probes corresponding to the highly expressed ESCs miRNAs miR-130–3 p, miR-293–3 p, and miR-294–3 p were synthesized using the mirVana miRNA Probe Construction Kit (Ambion) and hybridized overnight in 1xSSC, 1%SDS (w/v) and 100 mg/ml ssDNA. After hybridization, membranes were washed four times at 40°C in 0.2xSSC and 0.2%SDS (w/v) for 30 min each. Blots were analysed using a PhosphorImager (Molecular Dynamics) and ImageQuant TL software for quantification. Oligonucleotides used are listed in *Supplementary file 1*.

## Transfections of poly(I:C), DNA, miRNA mimics and Antagomirs

To activate the IFN response, cells were transfected with either the dsRNA analogue poly(I:C) (Invivogen) or the Y-shaped-DNA cGAS agonist (G3-YSD, Invivogen) using Lipofectamine 2000 (ThermoFisher). Transfections were performed in 24-well format, with cells approximately 80% confluent, using different concentrations of poly(I:C), from 0,5 to 2,5 µg per well (as indicated in the figures) or 0.5 µg of G3-YSD. Cells were incubated for approximately 16 hr for poly(I:C)- and 8 hr for DNA-transfections before harvest and further processing. IFN-β expression was measured using a quantitative ELISA kit (Mouse IFN-β, Quantikine, R and D systems) according to manufacturer's instructions. Cells were transfected with 2.5 µg/ml poly(I:C), incubated for 16 hr after which supernatant

was collected and assayed for IFN-β. To activate ESCs with exogenous IFN-β (R and D systems), cells were incubated with 10.000 U/ml of IFN-β for 4 hr, followed by RNA extraction and quantitative RT-PCR.

For the miRNA mimics (miScript, Qiagen) a final concentration of 1 µM was transfected into cells using Dharmafect (Dharmacon), incubated for the desired period and further processed. The same procedure was followed for the antagomirs (Dharmacon), but at a concentration of 100 nM. All experiments were performed in 24-well format, with cells at approximately 80% confluency.

## Quantitative RT-PCR

Total RNA from cells was isolated using Tri reagent (Sigma-Aldrich) according to the manufacturer's instructions. 0.5–1 µg RNA was subsequently reverse transcribed using M-MLV (Promega) and random hexamers, and used for quantitative PCR in a StepOnePlus real-time PCR machine (Thermo-Fisher) using GoTaq master mix (Promega). Data was analysed using the StepOne software package. Oligonucleotides used are listed in *Supplementary file 1*.

## Cell lysis and western blots

Cells used for Western blot analysis were lysed in RIPA buffer (50 mM TRIS-HCl, pH 7.4, 1% triton X-100, 0.5% Na-deoxycholate, 0.1% SDS, 150 mM NaCl, protease inhibitor cocktail (Roche), 5 mM NaF, 0.2 mM Sodium orthovanadate). Lysates were spun and protein concentrations measured using a BCA protein assay kit (BioVision). After adjusting protein concentrations, lysates were mixed with reducing agent (Novex, ThermoFisher) and LDS sample buffer (Novex, ThermoFisher) and boiled at 70°C for 10 min before loading on pre-made gels (NuPAGE, ThermoFisher). Proteins were transferred to nitrocellulose membrane using semi-dry transfer (iBlot2, ThermoFisher). Membranes were blocked for 1 hr at room temperature in PBS-T (0.1% Tween-20) and 5% milk powder before overnight incubation at 4°C with primary antibody. Antibodies used were: Anti-rabbit HRP (Cell Signaling Technology), Anti-mouse HRP (Bio-Rad), MAVS (E-6, Santa Cruz Biotechnology), PKR (ab45427, Abcam), MDA5 (D74E4, Cell Signaling Technology), RIG-I (D12G6, Cell Signaling Technology), phospho-IRF-3 (D601M, Cell Signaling Technology) and α-tubulin (CP06, Merck). Proteins bands were visualised using ECL (Pierce) on a Bio-Rad ChemiDoc imaging system. Protein bands were quantified using ImageJ (v1.51p) software and expression levels calculated normalized to α-tubulin.

## Luciferase assay

The 3'UTRs from *Mda5*, *Rig-I* and *Mavs* were amplified from genomic DNA based on the annotation from UTRdb (utrdb.ba.itb.cnr.it) using primers containing restriction sites. The fragments were cloned in the psiCHECK-2 vector (Promega) at the 3' end of the *hRluc* gene. Cells in 24-well format were transfected with 250 ng plasmid using Lipofectamine 2000 and incubated for 24 hr. Cells were subsequently lysed and assayed using the Dual-Glo Luciferase assay system (Promega). Luminescence was measured in a Varioskan flash (ThermoFisher) platereader.

## Proteomics

For the total proteome comparison, 6 replicates of the *Dgcr8*[-/-] and *Dgcr8*[resc] cell lines were prepared by lysing cells in Lysis buffer (50 mM TRIS-HCl, pH 7.4, 1% triton X-100, 0.5% Na-deoxycholate, 0.1% SDS, 150 mM NaCl, protease inhibitor cocktail (Roche), 5 mM NaF and 0.2 mM Sodium orthovanadate) at 4°C. Samples were subsequently sonicated 4 × 10 s, at 2µ amplitude, reduced by boiling with 10 mM DTT and centrifuged. The samples were further processed by Filter-aided sample preparation (FASP) by mixing each sample with 200 µl UA (8M Urea, 0.1 M Tris/HCl pH 8.5) in a Vivacon 500 filter column (30 kDa cut off, Sartorius VN01H22), centrifuged at 14.000 x g and washed twice with 200 µl UA. To alkylate the sample, 100 µl 50 mM iodoacetamide in UA was applied to the columns and incubated in the dark for 30 min, spun, followed by two washes with UA and another two washes with 50 mM ammonium bicarbonate. The samples were trypsinized on the column by the addition of 4 µg trypsin (ThermoFisher) in 40 µl 50 mM ammonium bicarbonate to the filter. Samples were incubated overnight in a wet chamber at 37°C and acidified by the addition of 5 µl 10% trifluoroacetic acid (TFA). The pH was checked by spotting onto pH paper, and peptide concentration estimated using a NanoDrop. C18 Stage tips were activated using 20 µl of methanol, equilibrated with 100 µl 0.1% TFA) and loaded with 10 µg peptide solution. After washing with 100 uL 0.1% TFA,

the bound peptides were eluted into a Protein LoBind 1.5 mL tube (Eppendorf) with 20 μl 80% acetonitrile, 0.1% TFA and concentrated to less than 4 μl in a vacuum concentrator. The final volume was adjusted to 6 μl with 0.1% TFA.

Five μg of peptides were injected onto a C18 packed emitter and eluted over a gradient of 2–80% ACN in 120 min, with 0.1% TFA throughout on a Dionex RSLnano. Eluting peptides were ionised at +2 kV before data-dependent analysis on a Thermo Q-Exactive Plus. MS1 was acquired with mz range 300–1650 and resolution 70,000, and top 12 ions were selected for fragmentation with normalised collision energy of 26, and an exclusion window of 30 s. MS2 were collected with resolution 17,500. The AGC targets for MS1 and MS2 were 3e6 and 5e4 respectively, and all spectra were acquired with one microscan and without lockmass. Finally, the data were analysed using MaxQuant (v 1.5.7.4) in conjunction with uniprot fasta database 2017_02, with match between runs (MS/MS not required), LFQ with one peptide required. Average expression levels were calculated for each protein and significant differences identified using a two tailed t-test assuming equal variance (homoscedasticity) with a p-value lower than 0.05.

## Stable cell lines overexpressing DGCR8, Dicer and MAVS

Plasmids containing the sequence of mouse DICER (pCAGEN-SBP-DICER1, Addgene), MAVS (GE-healthcare, MMM1013-202764911) and DGCR8 (*Macias et al., 2012*) were used to amplify the open reading frame using specific primers containing restriction sites (*Supplementary file 1*). The amplified and digested fragments were ligated in pLenti-GIII-EF1α for MAVS and pEF1α-IRES-dsRED-Express2 for DGCR8 and DICER. Verified plasmids containing the genes of interest were transfected in mESCs using Lipofectamine 2000 and selected with the appropriate antibiotic. After several weeks of selection, colonies were isolated, expanded and tested for expression by qRT-PCR and Western blot.

## Mitochondrial activity

The mitochondria specific dye Rhodamine 123 (Sigma-Aldrich) was used to measure mitochondrial activity. Suspended cells were incubated with Rhodamine 123 at 37°C and samples were taken at various intervals, washed three times with PBS at 4°C and the fluorescence measured in a VarioSkan flash (ThermoFisher) plate reader (excitation 508, emission 535).

## Inhibitors

Cells were pre-incubated with the inhibitors BX795, which blocks the phosphorylation of the kinases TBK1 and IKKε, and consequently IRF3 activation and IFN-β production (10 μM, Synkinase) and the inhibitor BMS345541, which targets IKβα, IKKα and IKKβ and consequently NF-κβ signalling (10 μM, Cayman Chemical) for 45 min before infection with TMEV. 24 hr post-infection in the presence of the inhibitor, infected wells were scored and the $TCID_{50}$ calculated. For Ruxolitinib (Cell Guidance Systems), cells were pre-incubated for 45 min with 50 μM Ruxotlitinib, infected with TMEV and incubated for 16 hr followed by extensive washing with PBS, RNA extraction and analysis by quantitative RT-PCR.

## CRISPR/Cas 9 targeting of mmu-miR-673

To create a cell line lacking mmu-mmiR-673–5p, the Alt-R CRISPR-Cas9 System (IDT) was used. Two different crRNAs were designed to target sequences within the pri-miRNA sequence hairpin to induce structural changes disrupting processing by the Microprocessor and DICER. Cas9 protein and tracrRNAs were transfected with the Neon Transfection System followed by cell sorting to create single cell clones. Genomic DNA was purified and screened by PCR followed by restriction site disruption analyses for the pri-miRNA sequence. Genomic DNA of the pri-miRNA sequence of candidates was amplified using primers in *Supplementary file 1*, and cloned into pGEMt-easy vector for sequencing.

## miRNA qRT-PCR

Total RNA (100 ng) was used to quantify mmu-mmiR-673–5p levels. RNA was first converted to cDNA using miRCURY LNA RT kit (Qiagen). cDNA was diluted 1/25 for RT-qPCR using miRCURY LNA SYBR Green kit and amplified using mmu-mmiR-673–5p specific primers (Qiagen) and U6 as a

loading control. Quantitative PCR was carried out on a Roche LC480 light cycler and analysed using the second derivative method.

## Data availability

All processed Mass spectrometry data is provided as *Figure 3—source data 1*, including LFQ intensity values for each protein detected in each of the samples. All raw data are available from corresponding author upon request.

## Acknowledgements

We thank our colleagues at the Institute of Immunology and Infection Research for advice and discussions. We thank JF Caceres, H Marks, R Zamoyska and Y Crow for critical reading of the manuscript, and R Blelloch, P Digard, E Gaunt and R Zamoyska labs for reagents. We thank Jimi Wills from the IGMM Mass spectrometry facility for analysis of protein samples. This work is supported by the Wellcome Trust (107665/Z/15/Z), LK is supported by a MRC-DTP in Precision Medicine Fellowship.

## Additional information

### Funding

| Funder | Grant reference number | Author |
| --- | --- | --- |
| Wellcome | 107665/Z/15/Z | Sara Macias |

The funders had no role in study design, data collection and interpretation, or the decision to submit the work for publication.

### Author contributions

Jeroen Witteveldt, Conceptualization, Data curation, Formal analysis, Supervision, Validation, Investigation, Methodology, Writing—original draft, Writing—review and editing; Lisanne I Knol, Formal analysis, Investigation, Writing—review and editing; Sara Macias, Conceptualization, Supervision, Funding acquisition, Investigation, Methodology, Writing—original draft, Writing—review and editing

### Author ORCIDs

Jeroen Witteveldt https://orcid.org/0000-0001-8247-7010
Lisanne I Knol https://orcid.org/0000-0002-8975-0398
Sara Macias https://orcid.org/0000-0002-0643-3494

### Decision letter and Author response

Decision letter https://doi.org/10.7554/eLife.44171.022
Author response https://doi.org/10.7554/eLife.44171.023

## Additional files

### Supplementary files

• Supplementary file 1. Sequences of the oligonucleotides used in this study.
DOI: https://doi.org/10.7554/eLife.44171.016
• Transparent reporting form
DOI: https://doi.org/10.7554/eLife.44171.017

### Data availability

All processed Mass spectrometry data is provided as Figure 3—source data 1, including LFQ intensity values for each protein detected in each of the samples. All raw data are available from corresponding author upon request.

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
