## [Decision Letter]

Thank you for sending your article entitled "MicroRNA-deficient embryonic stem cells acquire a functional Interferon response" for peer review at *eLife*. Your article has been evaluated by two peer reviewers, and the evaluation has been overseen by a Reviewing Editor and Tadatsugu Taniguchi as the Senior Editor.

Given the list of essential revisions, including some new experiments, the editors and reviewers invite you to respond within the next two weeks with an action plan and timetable for the completion of the additional work. In doing so, please describe point-by-point response to the issues raised by the reviewers.

We plan to share your responses with the reviewers and then issue a binding recommendation.

Essential revisions:

Biological relevance should be strengthened. Data showing the production of IFN protein and biological activities of the secreted IFN on ESC is needed, with or without virus infection in *Dgcr8^-/-^* ESCs. Furthermore, more mechanistic study is needed, for examples, whether the ESCs have abnormal epigenetic regulation of miR-673 and whether miR-673 expression was suppressed through ESC differentiation.

*Reviewer #1:*

Both human and murine ESCs are incapable to produce type I IFN in response to dsRNA, to avoid an unexpected responses to transponson derived transcripts. Previous studies suggested this attributed to a reduced expression of dsRNA receptors (TLR3, MDA5, RIG-I). The authors declared here that at least in murine ESCs, miRNAs take the central role to suppress IFN responses by targeting MAVS. By bioinformatic analysis and verifying experiments, they declared miR-673 was the exact functional microRNA targeting MAVS in mESCs. This is an interesting study, and adds potential new understanding in this area. However, the data are not strongly supportive of the conclusion, and there are some conclusions need modification or additional experimental evidence to support.

1) The main conclusion of the manuscript is "miRNA mediated IFN suppression is responsible for the reduced resistance of mESC to RNA virus". They draw this from the observation: i) *Dgcr8^-/-^* mESC were resistant to virus; ii) IFN mRNA were increased in *Dgcr8^-/-^* mESC. This is illogical, and correlation is not causation. MicroRNAs may target other anti-viral factors (e.g. ISGs directly) and cause reduced resistance to virus, along with IFN upregulation.

2) All through the paper, they only examined the IFNb by qPCR. As ESCs mildly response to IFNs, a solid data shown the production of IFN protein and biological activities of the secreted IFN on ESC is needed. Otherwise, they can't conclude that IFN was responsible for reduced virus infection in *Dgcr8^-/-^* ESCs.

3) They only observed that miRNA is responsible for resistance of IFN responses of mESCs. They still haven't explained the reason. Whether the ESCs have abnormal epigenetic regulation of miR-673? Which transcript factors caused this epigenetic change (*Oct4? Nanog*? etc.)? and through ESC differentiation, whether miR-673 expression was suppressed?

4) ESCs acquire IFN responses during differentiation. Does miR-673 have any activities on ESC colony formation, proliferation and differentiation, as miR-673 may regulated IFN response by regulating ESC differentiation indirectly, instead of directly on MAVS.

5) Genetic ablation of Dgcr8 or Dicer would influence many biological activities of ESCs, e.g. their differentiation. Whether the reduced virus in these ESCs were caused by abnormities in cell proliferation or apoptosis?

6) "Both inhibitors increased viral susceptibility in wild type cells lines, however, the effect was far greater in the knock out cell lines". We can't draw the conclusion from Figure 3A and Figure 3—figure supplement 1A. There lacks a vehicle control. In addition, western analysis of the activation of RIG-I/IFN pathways (pTBK1, pIRF3, pIKK, pIkB, pp65, etc.) would be more helpful.

7) Subsection “miR-673 is crucial to suppress antiviral immunity in ESCs”, first paragraph, they declare high expression of miR-185 and miR-673 in ESC, however, there were no data shown this.

8) Figure 4B, It's intriguing that overexpression is less effective that miRNA inhibition to increase the protein level of MAVS.

*Reviewer #2:*

In this manuscript Witteveldt et al. investigate the lack of innate immune response in mouse ESCs. They exquisitely demonstrate that a miRNA; miR-673 binds to the 3' UTR of MAVS and suppresses the IFN response. This is a very novel finding as it reveals the importance of the miRNA pathway regulating innate immunity in ESCs.

The experiments in this manuscript substantiate this conclusions and in my opinion there is no requirement for additional ones.

---

## [Author Response]

[Editors' note: the authors’ plan for revisions was approved and the authors made a formal revised submission.]

Reviewer #1:Both human and murine ESCs are incapable to produce type I IFN in response to dsRNA, to avoid an unexpected responses to transponson derived transcripts. Previous studies suggested this attributes to a reduced expression of dsRNA receptors (TLR3,MDA5,RIG-I). The authors declared here that at least in murine ESCs, miRNAs take the central role to suppress IFN responses by targeting MAVS. By bioinformatic analysis and verifying experiments, they declared miR-673 was the exact functional microRNA targeting MAVS in mESCs. This is an interesting study, and adds potential new understanding in this area. However, the data are not strongly supportive of the conclusion, and there are some conclusions need modification or additional experimental evidence to support.1) The main conclusion of the manuscript is "miRNA mediated IFN suppression is responsible for the reduced resistance of mESC to RNA virus". They draw this from the observation: i, Dgcr8^-/-^ mESC were resistant to virus; ii, IFN mRNA were increased in Dgcr8^-/-^ mESC. This is illogical, and correlation is not causation. MicroRNAs may target other anti-viral factors (e.g. ISGs directly) and cause reduced resistance to virus, along with IFN upregulation.

We agree that the global disruption of miRNA expression can potentially lead to changes in other cellular processes that might influence the antiviral response. However, our experiments clearly demonstrate that manipulating a single miRNA, using multiple approaches (miR-673 mimics, antagomirs or miR-673 KO cell lines) (Figure 5) or, directly the MAVS expression levels (Figure 4) is enough to recapitulate the phenotype observed in the absence of all miRNAs. Furthermore, Mass spectrometry and Western blot data show that MAVS is the only significantly upregulated factor in the absence of miRNAs, out of all proteins involved in sensing and activating IFN expression upon dsRNA stimulation, and including all detected ISGs.

Factors marked with a red asterisk are those that do not display differential expression in the absence of miRNAs, as shown by Western blot analyses (Figure 3—figure supplement 1). Those indicated with a blue asterisk are not differentially expressed in the absence of miRNAs, as determined by Mass spectrometry data analyses (Figure 3—source data 1), including those relevant ISGs that seem to confer antiviral protection in human ESCs^1^. MAVS (green asterisk) is the only factor in these pathways that is significantly upregulated in the absence of miRNAs, as shown by Mass spectrometry analyses, and confirmed by Western blot (Figure 3C).

Furthermore, TMEV and IAV are not targeted by miRNAs^2,3^, excluding any direct antiviral effects of miRNAs on these viruses. Based on all these data, we have concluded there is a negligible contribution of other antiviral factors in the cellular models studied here.

2) All through the paper, they only examined the IFNb by qPCR. As ESCs mildly response to IFNs, a solid data shown the production of IFN protein and biological activities of the secreted IFN on ESC is needed. Otherwise, they can't conclude that IFN was responsible for reduced virus infection in Dgcr8^-/-^ ESCs.

We agree that this is an interesting aspect that would benefit from further experimentation. We are currently measuring IFN-β protein production by ELISA, and comparing the response of ESCs with or without miRNAs to stimulation with IFN-β.

ELISA has been included as new Figure 2—figure supplement 2C. We are also including data using exogenous IFN-β stimulation in new Figure 2D. We have also blocked IFN signalling, these data have been included in new Figure 2C, Figure 2—figure supplement 2D, Figure 5F and Figure 5—figure supplement 3A.

3) They only observed that miRNA is responsible for resistance of IFN responses of mESCs. They still haven't explained the reason. Whether the ESCs have abnormal epigenetic regulation of miR-673? Which transcript factors caused this epigenetic change (Oct4? nanog? etc.)? and through ESC differentiation, whether miR-673 expression was suppressed?

We thank the reviewer for this suggestion. Indeed, miR-673 is expressed during pluripotency and it becomes silenced during differentiation (see new Figure 5G). Both ESCs lines used in this study were differentiated using retinoic acid, and miR-673 expression was assessed by qRT-PCR after 2 or 10 days of the differentiation protocol. Data shown in Figure 5G represents the average expression of miR-673-5p normalized to U6snRNA in both ESC lines. The mouse IFN-proficient cell line, NIH3T3, also silences miR-673 expression (Figure 5D). All these suggest that miR-673 expression and IFN-proficiency are incompatible.

We feel that studying the epigenetic regulation of miR-673 expression is beyond the scope of this manuscript, even though it is a very interesting aspect to follow-up. So far, there is no annotation for the promoter that drives expression of this miRNA, which challenges any approach to assess its epigenetic regulation. The miR-673 expression data during differentiation is included in new Figure 5G.

4) ESCs acquire IFN responses during differentiation. Does miR-673 have any activities on ESC colony formation, proliferation and differentiation, as miR-673 may regulated IFN response by regulating ESC differentiation indirectly, instead of directly on MAVS.

We thank the reviewer for highlighting this possibility. This is an aspect of miR-673 biology that we have already addressed. Transient reintroduction of miR-673 in miRNA-deficient ECSs does not significantly affect the relative expression of pluripotency markers *Nanog* and *Oct4* (see left graph in Author response image 2). More importantly, miR-673 knock-out cell lines also express similar levels of the pluripotency markers *Nanog* and *Oct-4* in comparison to wild-type ESCs, whereas in mouse fibroblasts (NIH3T3) they are absent as expected (see right graph in Author response image 2). These data suggest that differences in antiviral resistance upon miR-673 manipulation are not due to a spontaneous loss of pluripotency. We are currently measuring the proliferation capacity of the CRISPR cell lines (see point 5), in addition to providing pictures of these cultures to demonstrate their ability to form colonies and phenotypical similarities to the wild-type counterparts.

**Author response image 2. respfig2:** 

5) Genetic ablation of Dgcr8 or Dicer would influence many biological activities of ESCs, e.g. their differentiation. Whether the reduced virus in these ESCs were caused by abnormities in cell proliferation or apoptosis?

Disruption of miRNA expression by knocking-out *Dgcr8* or *Dicer* genesimpairs the ability of ESCs to differentiate^4,5^ and to efficiently transition from G1 to S phase of their cell cycle^6,7^. A genome-wide screening revealed that the reintroduction of only 14 single miRNAs rescued this phenotype^6^. These 14 miRNAs, including miR-294-3p or miR-295-3p, share a specific seed sequence (AAGUGCU) to silence *Cdkn1a* expression, thus allowing cell cycle progression. This seed sequence is not conserved in any of the miRNAs tested in this manuscript (miR-125a-5p, miR-125b-5p, miR-185-5p and miR-673-5p), suggesting that they may not be involved in the direct regulation of *Cdkn1a* level, or cell cycle progression (see Author response image 3).

**Author response image 3. respfig3:** 

We would also like to stress that infections in WT ESCs where miR-673 expression was knocked out by CRISPR or WT ESCs overexpressing MAVS produced a similar antiviral phenotype as the miRNA-deficient cells (*Dgcr8^-/-^*and *Dicer^-/-^*), suggesting that abnormalities in cell proliferation or apoptosis play an insignificant role in this regulation. However, to eliminate any remaining doubt of the function of miR-673-5p in cell cycle, proliferation analyses of miR-673^-/-^ cell lines are currently being performed.

6) "Both inhibitors increased viral susceptibility in wild type cells lines, however, the effect was far greater in the knock out cell lines". We can't draw the conclusion from Figure 3A and Figure 3—figure supplement 1A. There lacks a vehicle control. In addition, western analysis of the activation of RIG-I/IFN pathways (pTBK1, pIRF3, pIKK, pIkB, pp65, etc.) would be more helpful.

We apologize if these graphs were not clear. Both figures display values normalized to mock treated, which means ‘vehicle control treated’. Both inhibitors have been extensively tested before for their action and targets in several cell lines, as shown for BX795^8,9^ and BMS345541^10,11^. Controls for inhibitors are included as new Figure 3—figure supplement 1B and C.

7) Subsection “miR-673 is crucial to suppress antiviral immunity in ESCs”, first paragraph, they declare high expression of miR-185 and miR-673 in ESC, however, there were no data shown this.

We apologize for not including the references to previous studies showing the expression levels of both these miRNAs. Besides the data we show in Figure 5, early publications showed that both miR-185 and miR-673 are well expressed in mouse ESCs in a *Dicer* and *Dgcr8-*dependent manner^12,13^. Later publications confirmed the expression of both miRNAs in ESCs and also found that especially miR-673 decreased in expression upon differentiation^14,15,16,17^. Besides cell lines, both miRNAs are also readily detected in the early stages of embryonic development ^18,19,17^. New references have been added.

8) Figure 4B, It's intriguing that overexpression is less effective that miRNA inhibition to increase the protein level of MAVS.

Generating stable cell lines overexpressing proteins always result in variable expression levels, depending on insertion location and effect of the protein on the cell. This is one of the reasons why we have included two additional approaches to manipulate MAVS expression levels in ESCs and assess their effects on antiviral defence. Reintroduction of miRNAs targeting MAVS (Figure 5B), as well as, CRISPR KO cell lines for miR-673-5p expression, revealed consistent results with MAVS being a central factor switching on and off the IFN response in ESCs.

References:

1) Wu X, Dao Thi VL, Huang Y, et al. Intrinsic Immunity Shapes Viral Resistance of Stem Cells. Cell. 2018;172(3):423-438.e25. doi:10.1016/j.cell.2017.11.018.

2) Bogerd HP, Skalsky RL, Kennedy EM, et al. Replication of Many Human Viruses Is Refractory to Inhibition by Endogenous Cellular MicroRNAs. J Virol. 2014;88(14):8065-8076. doi:10.1128/JVI.00985-14.

3) De Cock A, Michiels T, De Cock A, Michiels T. Cellular microRNAs Repress Vesicular Stomatitis Virus but Not Theiler’s Virus Replication. Viruses. 2016;8(3):75. doi:10.3390/v8030075.

4) Wang Y, Medvid R, Melton C, Jaenisch R, Blelloch R. DGCR8 is essential for microRNA biogenesis and silencing of embryonic stem cell self-renewal. Nat Genet. 2007;39(3):380-385. doi:10.1038/ng1969.

5) Kanellopoulou C, Muljo SA, Kung AL, et al. Dicer-deficient mouse embryonic stem cells are defective in differentiation and centromeric silencing. Genes Dev. 2005;19(4):489-501. doi:10.1101/gad.1248505.

6) Wang Y, Baskerville S, Shenoy A, Babiarz JE, Baehner L, Blelloch R. Embryonic stem cell–specific microRNAs regulate the G1-S transition and promote rapid proliferation. Nat Genet. 2008;40(12):1478-1483. doi:10.1038/ng.250.

7) Murchison EP, Partridge JF, Tam OH, Cheloufi S, Hannon GJ. Characterization of Dicer-deficient murine embryonic stem cells. Proc Natl Acad Sci U S A. 2005;102(34):12135-12140. doi:10.1073/pnas.0505479102.

8) Feldman RI, Wu JM, Polok_off_ MA, et al. Novel Small Molecule Inhibitors of 3-Phosphoinositide-dependent Kinase-1. J Biol Chem. 2005;280(20):19867-19874. doi:10.1074/jbc.M501367200.

9) Clark K, Plater L, Peggie M, Cohen P. Use of the pharmacological inhibitor BX795 to study the regulation and physiological roles of TBK1 and IkappaB kinase epsilon: a distinct upstream kinase mediates Ser-172 phosphorylation and activation. J Biol Chem. 2009;284(21):14136-14146. doi:10.1074/jbc.M109.000414.

10) Burke JR, Pattoli MA, Gregor KR, et al. BMS-345541 is a highly selective inhibitor of I kappa B kinase that binds at an allosteric site of the enzyme and blocks NF-kappa B-dependent transcription in mice. J Biol Chem. 2003;278(3):1450-1456. doi:10.1074/jbc.M209677200.

11) Yang J, Amiri KI, Burke JR, Schmid JA, Richmond A. BMS-345541 Targets Inhibitor of B Kinase and Induces Apoptosis in Melanoma: Involvement of Nuclear Factor B and Mitochondria Pathways. Clin Cancer Res. 2006;12(3):950-960. doi:10.1158/1078-0432.CCR-05-1220.

12) Tang F. MicroRNA expression profiling of single whole embryonic stem cells. Nucleic Acids Res. 2006;34(2):e9-e9. doi:10.1093/nar/gnj009.

13) Babiarz JE, Ruby JG, Wang Y, Bartel DP, Blelloch R. Mouse ES cells express endogenous shRNAs, siRNAs, and other Microprocessor-independent, Dicer-dependent small RNAs. Genes Dev. 2008;22(20):2773-2785. doi:10.1101/gad.1705308.

14) Knelangen JM, van der Hoek MB, Kong W-C, et al. MicroRNA expression profile during adipogenic differentiation in mouse embryonic stem cells. Physiol Genomics. 2011;43:611-620. doi:10.1152/physiolgenomics.00116.2010.-Pluripotent.

15) Zhao B, Yang D, Jiang J, et al. Genome-wide mapping of miRNAs expressed in embryonic stem cells and pluripotent stem cells generated by different reprogramming strategies. BMC Genomics. 2014;15(1):488. doi:10.1186/1471-2164-15-488.

16) Hadjimichael C, Nikolaou C, Papamatheakis J, Kretsovali A. MicroRNAs for Fine-Tuning of Mouse Embryonic Stem Cell Fate Decision through Regulation of TGF-β Signaling. Stem Cell Reports. 2016;6(3):292-301. doi:10.1016/j.stemcr.2016.01.004.

17) Yang Q, Lin J, Liu M, et al. Highly sensitive sequencing reveals dynamic modifications and activities of small RNAs in mouse oocytes and early embryos. Sci Adv. 2016;2(6):e1501482. doi:10.1126/sciadv.1501482.

18) Chiang HR, Schoenfeld LW, Ruby JG, et al. Mammalian microRNAs: experimental evaluation of novel and previously annotated genes. Genes Dev. 2010;24(10):992-1009. doi:10.1101/gad.1884710.

19) Diez-Roux G, Banfi S, Sultan M, Geffers L, Anand S. A High-Resolution Anatomical Atlas of the Transcriptome in the Mouse Embryo. PLoS Biol. 2011;9(1):1000582. doi:10.1371/journal.pbio.1000582.